# CTLA4-Linked Autoimmunity in the Pathogenesis of Endometriosis and Related Infertility: A Systematic Review

**DOI:** 10.3390/ijms231810902

**Published:** 2022-09-18

**Authors:** Mislav Mikuš, Marina Šprem Goldštajn, Ivan Brlečić, Stipe Dumančić, Antonio Simone Laganà, Vito Chiantera, Goran Vujić, Mario Ćorić

**Affiliations:** 1Department of Obstetrics and Gynecology, Clinical Hospital Center Zagreb, 10000 Zagreb, Croatia; 2Department of Obstetrics and Gynecology, Clinical Hospital Center Sestre Milosrdnice, 10000 Zagreb, Croatia; 3Department of Obstetrics and Gynecology, Clinical Hospital Center Split, 21000 Split, Croatia; 4Unit of Gynecologic Oncology, ARNAS “Civico-Di Cristina-Benfratelli”, Department of Health Promotion, Mother and Child Care, Internal Medicine and Medical Specialties (PROMISE), University of Palermo, 90127 Palermo, Italy

**Keywords:** endometriosis, infertility, CTLA4, reproductive immunology, pathophysiology, systematic review

## Abstract

Several studies, although with conflicting results, have sought to determine the concentration of soluble CTLA4 antigens in peripheral blood plasma and peritoneal fluid in patients with endometriosis-related infertility. A systematic review was performed according to the Preferred Reporting Items for Systematic Reviews and Meta-Analyses (PRISMA) through a search of the following databases: MEDLINE, EMBASE, Global Health, The Cochrane Library, Health Technology Assessment Database and Web of Science, and Clinical Trials research register. We included observational or prospective human and animal studies with any features related to endometriosis and/or infertility studies involving CTLA4-related pathogenesis published in English. The results of studies in which the size and characteristics of the observed groups were not stated were excluded. From the initial pool of 73 publications identified and screened, we finally included 5 articles to summarize the most recent knowledge about CTLA4-linked autoimmunity in the pathogenesis of endometriosis and related infertility. Evidence from clinical studies shows that CTLA4-based autoimmunity is involved in the maintenance of chronic inflammation in the peritoneal environment, with pre-clinical evidence of anti-CTLA antibodies as a potential novel target therapy for endometriosis. However, CTLA4 gene analyses do not support findings of CTLA4-linked autoimmunity as a primary determinant of the pathogenesis of endometriosis. These findings underlie the role of complex interactions within the family of immune checkpoint molecules involved. Further studies are needed to investigate the clinical relevance of anti-CTLA target therapy, taking into account the potential adverse events and repercussions of novel immunologic therapy modalities. However, with the general scarcity of studies investigating this topic, the clinical importance of CTLA4 autoimmunity still remains unclear.

## 1. Introduction

Endometriosis presents one of the most common chronic gynecological conditions associated with cyclic pelvic pain, subfertility or both [1,2]. It affects approximately 20% of women hospitalized for pelvic pain and is associated with infertility in 50% of these cases [3,4]. The symptoms of endometriosis vary from mild to severe, but in most cases, women suffering from endometriosis need an important monthly intake of analgesics and generally have a poor health-related quality of life [5,6,7]. Furthermore, the significant diagnostic delay of endometriosis, as well as the search for effective pharmacological strategies, causes a substantial economic burden [8]. Women with endometriosis are also at an increased risk of developing several cancers [9,10,11] and autoimmune disorders.

The exact pathogenesis of endometriosis remains elusive, but a number of theories have been proposed that integrate several genetic and epigenetic theories [12]. More recent work emphasizes the importance of dysregulation of the immune system in the progression of the disease [13,14]. Thus, the widely accepted Sampson’s theory can be applied to the majority of women, but it has been hypothesized that endometriosis develops in women whose dysregulated immune mechanisms cannot overcome the appropriate response to refluxing endometrial debris [15]. In the last decade, studies on apoptosis [16,17,18] and immune checkpoint molecules [19,20] have focused on reproductive immunology. Those molecules act as inhibitory signaling mediators to maintain immune tolerance by several mechanisms: regulation of T-cell homeostasis, promoting T regulatory (Treg) cell development, inhibition of autoreactive T cells and inhibition of the effector T-cell differentiation and cytokine production leading to immunosuppression [21,22]. One of the immune checkpoint molecules, cytotoxic T lymphocyte–associated antigen-4 (CTLA4), particularly its soluble form (sCTLA4), has been correlated with several autoimmune diseases, such as autoimmune thyroid disease, type 1 diabetes mellitus, and immune thrombocytopenia and has been linked with endometriosis-related infertility [23]. Inhibitory in its nature, CTLA4 is a critical immunoregulatory molecule that belongs to the family of type I membrane receptors [24]. CTLA4 is an important structure in signaling between cells of the immune system, downregulating T-cell activation and favoring the anergic state of lymphocytes [25].

Several studies, although with conflicting results, have sought to determine the concentration of the sCTLA4 antigen in peripheral blood plasma and peritoneal fluid in patients with endometriosis-related infertility [23,26,27,28,29]. Data suggest that significantly higher percentages of CD4^+^/CTLA4 and CD8^+^/CTLA4 T lymphocytes have been observed in patients with endometriosis and intraoperative adhesions, implicating that there is a possible correlation between the ratio of CD8^+^ T/CD4^+^ T and endometriosis-related infertility [28]. These studies have expanded the general knowledge of the pathophysiology of endometriosis and may open up new potential diagnostic and treatment options for endometriosis-related infertility. Considering the increasing attention to the problem, in this article, we aimed to summarize the most recent knowledge about the topic.

## 2. Materials and Methods

A systematic review was performed according to the Preferred Reporting Items for Systematic Reviews and Meta-Analyses (PRISMA) [30], through a search of the following databases: MEDLINE, EMBASE, Global Health, The Cochrane Library (Cochrane Database of Systematic Reviews, Cochrane Central Register of Controlled Trials, Cochrane Methodology Register), Health Technology Assessment Database and Web of Science, and Clinical Trials research register. The aim was to summarize data from relevant articles regarding the correlation between CTLA4 and endometriosis-related infertility. The search included the following keywords, from the inception of respective databases to 25 July 2022: CTLA4, anti-CTLA4 antibody, endometriosis, autoimmunity, pathogenesis, female infertility and immune therapy. EndNote 20 reference manager was used to combine and remove duplicate results. Three authors (M.M., S.D. and A.S.L.) independently screened the titles and abstracts of the studies obtained by the search strategy. The full text of each potentially relevant study was obtained and assessed for inclusion independently by the two authors (I.B. and G.V.). They also independently extracted data from the included studies using a pre-piloted standard form to ensure consistency of the data extraction. Three other authors (M.Š.G., V.C. and M.Ć.) independently reviewed the selection and data extraction processes.

We included observational or prospective human and animal studies with any features related to endometriosis and/or infertility studies involving CTLA4-related pathogenesis published in English. The results of studies in which the size and characteristics of the observed groups were not stated were excluded. Studies without original data, including reviews, comments, editorials and meta-analyses, were excluded (Appendix A). The results were compared, and any disagreement was discussed and resolved by consensus. Studies providing ambiguous or insufficient, low-quality data or not quantifiable outcomes were also excluded. Due to the nature of the findings, we opted for a narrative synthesis of the results from the selected articles.

## 3. Results

### 3.1. Literature Search

The electronic searches, after duplicate records were removed, provided a total of 73 citations. Of these, 68 were excluded after title/abstract screening (not relevant to the review inclusion criteria). We examined the full text of 5 publications remaining to summarize the most recent knowledge about CTLA4-linked autoimmunity in the pathogenesis of endometriosis and related infertility [23,26,27,28,31]. The PRISMA flow chart of the literature search is reported in Figure 1.

### 3.2. Study Characteristics

Table 1 shows the detailed characteristics of the included studies after the literature search [23,26,27,28,31]. Only one study has been conducted on animal models, specifically, in the peritoneal fluid of a mouse endometriosis model [31], while other four studies were case-control studies comparing either infertile women with or without endometriosis or women with histologically confirmed endometriosis and women without any sign of disease [23,26,27,28].

## 4. Discussion

### 4.1. Membrane-Bound and Soluble CTLA4 Antigen Involved in Endometriosis and/or Infertility

CTLA4 antigen is found in two forms: membrane-bound protein CTLA4 and free, sCTLA4 form found in serum and peritoneal fluid [32]. The latter is considered one of the immune checkpoint molecules with roles in carcinogenesis, inflammation and pregnancy [23]. Membrane-bound CTLA4 directly competes with CD28 by binding to the same ligands, CD80 (B7.1) and CD86 (B7.2), and thus is one of the inhibitory regulators of lymphocyte activation [33].

The role of CTLA4 in endometriosis pathophysiology is summarized in Figure 2. Abramiuk et al. [28] aimed to evaluate the expression of CTLA4 on the surface of CD4^+^, CD8^+^ and B19^+^ lymphocytes and to determine the concentration of sCTLA4 antigen in serum and peritoneal fluid in patients. A similar study was conducted by Santoso and colleagues [23], who hypothesized that soluble immune checkpoint molecules are elevated during endometriosis in infertile women in peripheral serum and peritoneal fluid. CD8^+^ T cells without CTLA4 expression result in significantly higher IFN-gamma and granzyme B production and enhanced cytolysis in an animal model, which can be in concordance with significantly higher CTLA4 antigen expression among CTLA4 T cells in the serum of patients with endometriosis, with a positive correlation of the percentage of CD4^+^ and CD8^+^ T cells and the endometriosis stage [28]. A recent study also showed that there is a higher Treg cell percentage in the serum of patients with endometriosis, without similar findings in peritoneal fluid [23]. Moreover, Abramiuk et al. [28] showed the role of immunosuppressive mechanisms in the development of endometriosis-related infertility: CD4^+^ and CD8^+^ T lymphocytes negatively correlated with the percentage of NK and NKT-like cells (confirming previous findings [34]), while CD8^+^ T cells positively correlated with the percentage of CD4^+^CD25^+^Foxp3^high^ Treg cells.

Santoso and coworkers found that immune checkpoint molecules, with involvement of sCTLA4 in the peritoneum, act as immune regulators in the pathogenesis of endometriosis [23]. They found significantly higher serum sCTLA4 in the late-stages (III and IV) compared to early-stages (I and II) endometriosis and the control group, which can indicate the peripheral tolerogenic role of sCTLA4 on immune function in endometriosis patients. Furthermore, higher levels of sCTLA4 in the serum of advanced-stage endometriosis might be the outcome of increased proteolytic cleavage of membrane forms from local endometrial-like cell lesions, which then circulate into the periphery. This study also found higher concentrations of sCTLA4 in the peritoneal fluid compared to their concentration in serum, as well in the peritoneal fluid of women with endometriosis-related infertility compared to their gynecologic control [23]. Finally, serum and peritoneal concentrations of sCTLA4 were positively correlated, which can indicate that serum levels originate from the local peritoneal cavity.

### 4.2. CTLA4 Gene Variants in the Pathogenesis of Endometriosis

CTLA4 gene, mapped on the 2q33 chromosome, is associated with susceptibility to autoimmunity, with the CTLA4 region as an important locus for autoimmune diseases [35]. For this reason, several genetic polymorphisms of CTLA4 gene, with prominent CTLA4+49A/G polymorphism (exon 1) and CT60 A/G dimorphism (3′-UTR), were already investigated [36].

Two research groups [26,27] conducted case-control studies to investigate the genotype frequencies of the latter variants in patients with endometriosis compared to healthy controls. Both groups found that CTLA4 gene variants do not play a role in the pathogenesis of endometriosis, with or without related female infertility. Viganò et al. formed two hypotheses of autoimmune etiology of endometriosis [26]; on the one hand, autoimmunity can underlie endometriosis development without CTLA4-dependent predisposition; on the other hand, endometriosis is not associated with autoimmune etiology but can predispose the development of autoimmune reactions. Nevertheless, further studies are needed to investigate genotype frequencies between different severity and manifestations of endometriosis. Indeed, Viganò et al. did not find any association of the variants’ distribution with parameters indicative of a specific manifestation of the disease [26]. However, Lerner et al. [27] also found no differences in the frequency of CTLA4+49A/G polymorphism in patients with minimal-to-mild and moderate-to-severe endometriosis compared to controls. These findings may underlie the fact that endometriosis etiology, irrespective of related infertility, is not primarily associated with the development of CTLA4-related autoimmunity.

### 4.3. Anti-CTLA4 Antibodies as Target Therapy for Endometriosis?

Although there are substantial improvements in hormonal and non-hormonal therapies [37,38,39], majority of the currently available treatment options for endometriosis suppress ovarian function, are partially or totally contraceptive, and do not present a final solution for patients [40]. Accumulating evidence suggests that immunotherapy may open new scenarios for the treatment of refractory and recurrent cases of endometriosis [41]. Liu et al. conducted an experimental study with endometriosis mouse model [31]. They hypothesized that PLGA/anti-CTLA4 suppress ectopic endometrial cell proliferation and invasion by reducing IL-10 and TGF-β production secreted by CD4^+^CD25^+^Treg cells in a sustained-release manner [31].

Pathological immune response in the peritoneal environment is responsible for the implantation of ectopic endometrium [42] and maintenance of immunological self-tolerance mediated by CD4^+^CD25^+^ Treg cells, which were found in the peritoneal fluid of women with endometriosis in higher concentrations. The latter subpopulation of T cells is activated via CTLA4-modified binding of CD80/CD86 ligands to the TCR of T cells, which can stimulate endometrial stromal cell proliferation and invasion by secreted suppressive cytokines IL-10 and TGF-β (Figure 2).

Liu et al. formulated a drug delivery system of anti-CTLA4 antibody encapsulated with polylactic-co-glycolic acid (PLGA), which ensured sustained release of the antibody in an in vitro micro-environment [31]. They found that both proliferation and invasion of ectopic endometrial cells were suppressed with PLGA/anti-CTLA4, reducing IL-10 and TGF-β secreted by CD4^+^CD25^+^Treg cells. Thus, this study provided preliminary experimental results for anti-CTLA-based therapy for endometriosis. However, treatment with the anti-CTLA4 antibody results in a broad spectrum of adverse events, at least in the investigated mouse model, such as the suppressive influence on Tregs. Although this finding from the mouse model cannot be translated directly into humans, it is possible that the anti-CTLA4 antibody may exert the same suppressive effect on Tregs in humans, leading to potential severe side effects due to reduction of self-tolerance and thus increased autoimmunity [43].

## 5. Conclusions

This systematic review provides a summary of the available pieces of evidence about CTLA4-based autoimmunity in the pathogenesis of endometriosis and related infertility. Data from clinical studies show that CTLA4-based autoimmunity is involved in the maintenance of chronic inflammation in the peritoneal environment, with pre-clinical evidence of anti-CTLA antibodies as a potential novel target therapy for endometriosis. However, CTLA4 gene analyses do not support findings of CTLA4-linked autoimmunity as a primary determinant of the pathogenesis of endometriosis. These findings underlie the role of complex interactions within the family of immune checkpoint molecules involved. Further studies are needed to investigate the clinical relevance of anti-CTLA target therapy, taking into account the potential adverse events and repercussions of novel immunologic therapy modalities. However, due to the low number of studies investigating this topic, the clinical importance of CTLA4 autoimmunity in endometriosis remains unclear.

## Figures and Tables

**Figure 1 ijms-23-10902-f001:**
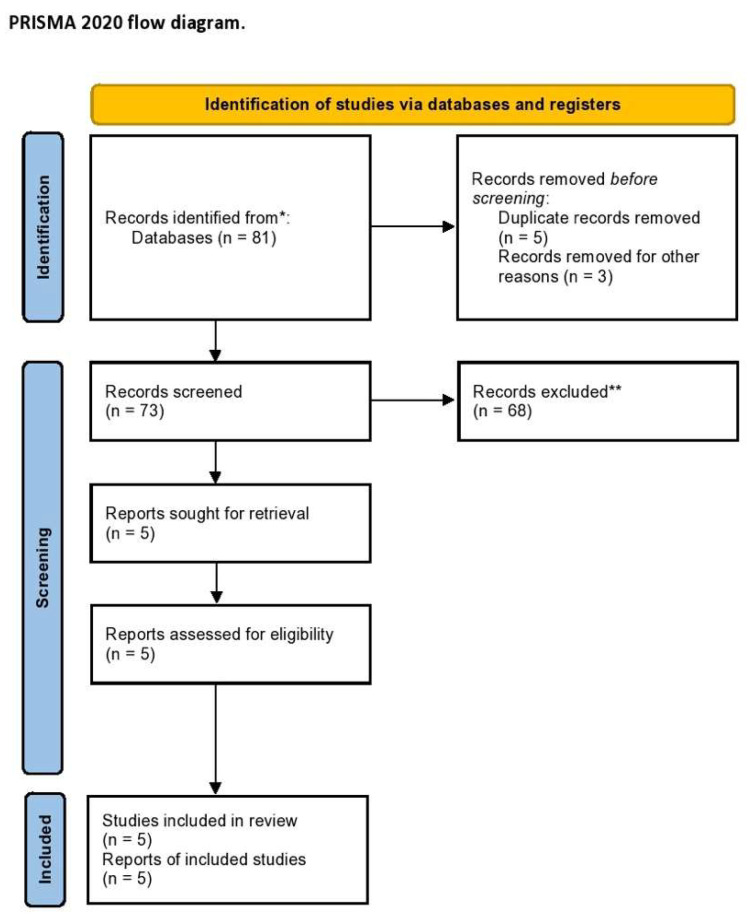
PRISMA flow diagram of the systematic literature search. * MEDLINE, EMBASE, Global Health, The Cochrane Library (Cochrane Database of Systematic Reviews, Cochrane Central Register of Controlled Trials, Cochrane Methodology Register), Health Technology Assessment Database and Web of Science, and Clinical Trials; ** Records excluded by a human and automation tools.

**Figure 2 ijms-23-10902-f002:**
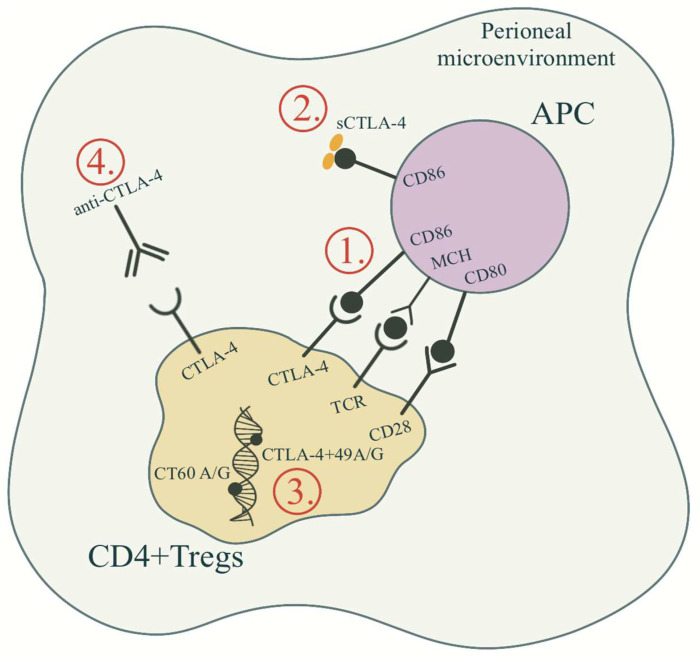
The mechanisms of CTLA4 involvement in the pathogenesis of endometriosis. (1) The CTLA4 expressed on the surface of T cells involved in inhibitory co-stimulatory signal in T cell proliferation; (2) Higher serum and peritoneal soluble form of CTLA4 (sCTLA4) found in advanced endometriosis and related infertility; (3) CTLA4 gene polymorphisms not associated with the pathogenesis of endometriosis; (4) Anti-CTLA4 antibodies in mouse endometriosis model found to reduce the number of Treg cells and thus inhibit proliferation and invasion of ectopic endometrium. (Original illustration).

**Table 1 ijms-23-10902-t001:** Characteristics of the included studies.

Study	Study Type	Study Population		Results
		Cases	Controls	
Viganò et al. (2005) [26]	Case-control study	150 women ^1^	168 women ^2^	No difference in genotype frequencies of CTLA4^+^ 49A/G and CT60A/G between endometriosis patients and controls Frequency of the G alleles of CTLA4^+^ 49A/G and CT60A/G genotypes is not associated with parameters of endometriosis manifestation or severity ^a^
Lerner et al. (2011) [27]	Case-control study	244 infertile women 177 endometriosis-related ^1^ 67 idiopathic ^2^	172 fertile women ^2^	No difference in genotype frequencies of CTLA4^+^ 49A/G between patients with minimal-mild and moderate-severe endometriosis No association of CTLA4^+^ 49A/G polymorphisms and endometriosis-related or idiopathic infertility compared to controls ^a^
Santoso et al. (2020) [23]	Case-control study	44 infertile women ^1^	44 infertile women ^2^	Higher serum sCTLA4 in late-stage endometriosis compared to early-stage and control groups Higher peritoneal sCTLA4 in endometriosis-related infertility compared to gynecologic control Higher sCTLA4 in late-stage endometriosis compared to the control Positive correlations between serum and peritoneal sCTLA4 concentrations Higher serum and peritoneal sCTLA4 with late-stage endometriosis compared to fallopian- and myoma-related infertility without endometriosis sCTLA4 showed lower diagnostic value (sensitivity 70.45%, specificity 55.17%; cut-off concentration 75.53 pg/mL) Serum sCTLA4 in correlation with EFI score, rASRM endometriosis score and dysmenorrheal symptoms
Abramiuk et al. (2021) [28]	Case-control study	54 women ^1^	20 women ^2^	Higher CTLA4 expression on serum CD8^+^ T cells with endometriosis compared to controls Positive correlation between endometriosis stage and percentage of CD4^+^ T and CD8^+^ T cells Endometriosis-related infertility: negative correlation between number of CD4^+^T and percentage of NKT and NKT-like CD3^+^CD16^+^CD56^+^ cells, as well as CD8^+^ T; positive correlation of number of CD8^+^ T and the percentage of Treg CD4^+^CD25^+^Foxp3^high^ Endometriosis and intraoperative adhesions: higher percentages of CD4^+^CTLA4 and CD8^+^CTLA4 T cells Differences in serum sCTLA4 concentration between endometriosis and control group
Liu et al. (2016) [31]	Animal in vitro study	30 mouse endometriosis models (LPTM transplanted with autologous uterine endometrium)	/	Anti-CTLA4 released from PLGA/anti-CTLA4 microspheres sustained a period of long time even in day 24 PLGA/anti-CTLA4, when added to the culture system with ectopic endometrial cells (isolated from the peritoneal fluid of the mouse endometrial model), inhibited proliferation and invasion of endometrial stromal cells through reduced IL-10 and TGF-β secreted by CD4^+^CD25^+^Treg cells

^1^ Women with endometriosis diagnosed by laparoscopy and confirmed by histopathological examination. ^2^ Women without signs of endometriosis observed by laparoscopy; ^a^ Observed by ORs—calculated using allele frequencies and G variants of the polymorphisms as risk factors for the disease; *(s)CTLA4* (soluble) cytotoxic T-lymphocyte antigen 4, *EFI* Endometriosis Fertility Index, *rASRM* revised American Society for Reproductive Medicine score, *ORs* odds ratios, *LPTM* laparotomy, *PGLA* poly(lactic-co-glycolic acid).

## Data Availability

Not applicable.

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
