# Peer review of "CTLA4-Linked Autoimmunity in the Pathogenesis of Endometriosis and Related Infertility: A Systematic Review"

_ijms, 2022, doi:10.3390/ijms231810902_

Round 1
Reviewer 1 Report
Endometriosis is an inflammatory, estrogen dependent condition associated with pelvic pain and infertility. Although the pathophysiology and pathogenesis of the condition is not fully described, Sampson’s theory of retrograde menstruation is a widely accepted cause of endometriosis. In recent years, many studies have implicated the involvement of CTLA4 in endometriosis. The manuscript tries to combine results from all studies linking CTLA4 and endometriosis that has been published till date through a comprehensive search of various databases. Based on results from 5 publications, the authors then try to determine if CTLA4 is linked to endometriosis or not.
The Materials and Methods section is well described, and the approach is acceptable for the kind of study presented in this article.
The manuscript has lot of typographical errors which is listed below in Minor Comments and should be corrected.
Major Comments
1. Results section: Of the 73 articles obtained from literature search, 68 were excluded because it did not meet review inclusion criteria. Although it is mentioned about exclusion criteria in the Methods section, it’d be helpful to include a table in which the Pubmed ID’s of all excluded 68 articles and the reason for their exclusion listed. This will help the reader understand what the shortcoming of the excluded studies were in respect to the manuscript being presented here. This can be presented as a table like Table1, but in a summarized form.
2. Also, list the 5 articles that met the inclusion criteria in the same table and describe how they met the inclusion criteria.
3. Fig1: “Reports not retrieved (n=5)”, what does this mean?
4. Please list some of the adverse events observed in the mouse model in response to anti-CTLA4 antibody treatment.
Minor Comments
1. Introduction, Para1: Correct as “Furthermore, the significant diagnostic delay of endometriosis as well as the search of effective pharmacological”
2. Introduction, Para2: correlated misspelled ad colerrelated.
3. Materials and Methods, Para1: Correct as “The aim was to summarize data”
4. Materials and Methods, Para1: Correct as “from the inception of respective databases”
5. Discussion, Para2: Correct as Abramiuk et al.
6. Fig 2 legend: involved misspelled as inovolved.
Reviewer 2 Report
Dear Colleagues,
Thank you very much for submitting your manuscript. Endometriosis is a frequent gynaecological pathology with significant implications for medical practice. The topic is relevant and exciting to the field of the journal. The text is clear and easy to read. The manuscript has an excellent structure and description. The overall paper is organized well written. The literature reviews are insightful and informative.
The figures are well presented and easy to read and understand. The presented aspects sufficiently support the conclusions.
I congratulate all the authors for their efforts.
I have only a few remarks to make:
- fine/minor spell check required
- the references are double numbered
Reviewer 3 Report
Absctract:
Please, add the methodology outline in the abstract:
How many articles were preliminary included in database [precise databases]?
What the inclusion and exclusion criteria were?
"..., we designed and performed ..." - it seems a bit trivial
Introduction:
In wchich diseases/s sCTLA-4 is the highest? Are these disease related to infertility?
Is the ratio of CD8+ T/CD4+ T related to sCTLA-4 and infertility?
Materials and methods:
too low number of the original papers to write good review article
line 6: " Thea im was to summarize" => The aim was...?
Results:
Figure 1 has too large white margin.
Table 1 needs visual correction.
Table 1: "... inhibited proliferation and invasion of cells through reduced IL10 and TGF-beta" - which cells? Is mouse suplementation with PLGA/antiCTLA-4 related with mouse infertility? (this is further described in Discussion, but should be also included in table)
Discussion:
line 13: INF-g => INF-gamma
line 16 and 17: + => +
Figure 2: to large white area
Conclusions:
"However, with the general scarcity of studies investigating
this topic, the clinical importance of CTLA4 autoimmunity in endometriosis remains unclear." - is like: "I know that I know nothing" - so what is the point of reading this article to still know nothing
Suggestions:
Consider to add the abbreviations list, please.
Unify CTLA4 or CTLA-4, please.
Round 2
Reviewer 3 Report
I thank the authors for corrections and answers. In most importance, the list of excluded articles in supplementary material and the number of current references explain in some way the low number of included articles in this review and shows the exact work.
Author Response
We are sorry for omitting Appendix Table from the main document (it was just an oversight, we actually uploaded the table wrongly as supplementary material and we did the job in the previous round of review). Thank you very much for your review and acknowledgments.